# Breakthrough Tick-Borne Encephalitis and Epidemiological Trends in an Endemic Region in Poland: A Retrospective Hospital-Based Study, 1988–2020

**DOI:** 10.3390/vaccines13070665

**Published:** 2025-06-20

**Authors:** Magdalena Sulik-Wakulińska, Kacper Toczyłowski, Sambor Grygorczuk

**Affiliations:** 1Department of Infectious Diseases and Neuroinfections, Medical University in Białystok, Żurawia 14, 15-540 Białystok, Poland; sambor.grygorczuk@umb.edu.pl; 2Department of Pediatric Infectious Diseases, Medical University in Białystok, Waszyngtona 17, 15-274 Białystok, Poland; kacper.toczylowski@umb.edu.pl

**Keywords:** tick-borne encephalitis, vaccine effectiveness, breakthrough infection

## Abstract

**Background/Objectives**: Tick-borne encephalitis (TBE) is a notifiable disease in Poland, with the highest incidence in the northeastern region. Although vaccination is highly effective, breakthrough infections occasionally occur. This study aimed to describe the clinical features of vaccinated and unvaccinated TBE cases, assess long-term hospitalization trends, and estimate vaccine effectiveness (VE) in a highly endemic region. **Methods**: We retrospectively analyzed 1518 laboratory-confirmed TBE cases hospitalized at the University Clinical Hospital in Białystok, Poland, from 1988 to 2020. Clinical and cerebrospinal fluid (CSF) parameters were compared between vaccinated and unvaccinated individuals. Vaccine effectiveness was estimated using the screening method, based on aggregated regional vaccine uptake data from 1999 to 2020. **Results**: Among all cases, 13 (0.9%) occurred in individuals who had received at least one dose of vaccine, including 4 who had completed the full primary vaccination schedule. Hospitalized vaccinated patients showed similar demographic and clinical characteristics compared to unvaccinated patients, though CSF findings suggested an earlier and more dynamic immune response. Seasonal analysis revealed a sustained increase in TBE hospitalizations and a possible extension of the transmission season into late summer and autumn. Estimated VE was 94.4% (95% CI 85.2–97.9%), though this should be interpreted with caution due to the small number of vaccinated cases and assumptions regarding population-level coverage. **Conclusions**: This study provides detailed clinical data on breakthrough TBE cases and long-term epidemiological insights from an endemic region in Poland. While vaccine effectiveness appears high, low uptake remains a public health concern. These findings underscore the need for improved vaccination coverage and ongoing surveillance to monitor evolving transmission patterns.

## 1. Introduction

Tick-borne encephalitis virus (TBEV) is a zoonotic pathogen of the genus Flavivirus and the causative agent of tick-borne encephalitis (TBE), a potentially severe central nervous system disease. TBE is endemic across much of Eurasia, with three main subtypes (European, Siberian, and Far Eastern), and has been expanding into new regions of Europe in recent years [1,2,3]. Transmission to humans occurs mainly via tick bites, and while infection can be asymptomatic, symptomatic cases may result in serious neurological complications or death. The majority of infections are asymptomatic, but the true prevalence of subclinical cases varies by population. Symptomatic TBE typically presents as a biphasic illness, with some patients developing meningitis, meningoencephalitis, or meningoencephalomyelitis [4].

There is no specific antiviral treatment for TBE, making prevention through vaccination particularly important. Two licensed vaccines and their pediatric formulations are available in Europe and have demonstrated high effectiveness when administered according to the recommended schedules. Long-term immunity studies indicate that antibody persistence can extend up to 15 years after a booster dose [5,6]. Vaccine effectiveness is estimated at approximately 96.6% for individuals who complete the three-dose primary series with timely boosters. Breakthrough infections are rare and are often associated with incomplete vaccination schedules or occur in older adults, possibly due to age-related immune decline rather than vaccine failure [5]. When breakthrough infections do occur, the course is generally less severe than in unvaccinated individuals, although severe outcomes remain possible and may be influenced by host factors rather than vaccine limitations. Antibody-dependent enhancement, a phenomenon seen in some flaviviral infections, does not appear to play a significant role in TBE [5].

Despite the high effectiveness of vaccination, data on the clinical course and outcomes of TBE in large patient cohorts remain limited. This study aims to provide long-term clinical observations from a highly endemic region in Poland, focusing on the clinical course, laboratory findings, and long-term outcomes of TBE, as well as the characteristics and severity of vaccine breakthrough infections and the overall effectiveness of TBE vaccines.

## 2. Materials and Methods

### 2.1. Study Design and Population

We conducted a retrospective study of patients with laboratory-confirmed TBE who were hospitalized at the University Clinical Hospital in Białystok, Poland, between 1988 and 2020. Bialystok is located in the Podlaskie Voivodship in northeastern Poland, a region endemic for TBE.

### 2.2. Definition of TBE

TBE is a notifiable disease in Poland, occurring mostly in the studied region. Defined cases followed the national notification definition [7]. In this study, TBE cases are defined as laboratory-confirmed infections with clinical signs of CNS involvement, in line with national reporting standards. As only hospitalized patients with CNS symptoms were included, our findings represent the more severe end of the disease spectrum. TBE was confirmed in all patients by using an enzyme-linked immunosorbent assay for detection of immunoglobulin (Ig)M and IgG antibodies in serum and/or CSF, tested with the routine diagnostic method used at the time.

### 2.3. Data Collection

We included all the patients with TBE treated in University Clinical Hospital in Białystok from 1988 to 2020. Characteristics of age, gender, and number of vaccine doses, as well as clinical and laboratory data, were obtained from medical records. An appraisal of the severity of clinical disease in the acute phase was performed. In our analysis, we included the assessment of CSF. The date of CSF collection was also included in the analysis. Most patients had their first lumbar puncture performed within the first two days of admission. Then, some of the patients had additional second tests after 8–25 days. Few people had their third puncture performed at least 26 days after hospitalization. The results of the cerebrospinal fluid test in both groups were compared over time. The outcome is presented by means of a graph.

### 2.4. Definitions of Vaccination Status

The standard vaccination schedule was initially 2 primary doses (at 0 and 1–3 months), followed by a third dose at 5–12 months, and then first booster at 3 years. Furthermore, booster doses were recommended every 5 years for majority of people or every 3 years for all individuals aged ≥60 years [8]. Previously vaccinated patients included various individuals. The ones that adhered to the recommended schedule in Poland during a specific year and were boosted, if necessary, were called completely vaccinated and protected. The other ones, who did not comply with the schedule and did not receive a full course of vaccines (received 1–2 doses only), were termed incompletely vaccinated. Patients who received a full course of a primary schedule but skipped their booster doses were termed “fully vaccinated, not boosted”.

### 2.5. Statistical Analysis

Statistical analyses were performed using a two-sided alpha level of 0.05. Since none of the clinical parameters followed a normal distribution, numeric variables were summarized as medians (*Mdn*) and interquartile ranges (IQR). Comparisons of these variables between two independent groups were conducted using the Wilcoxon rank sum test. For categorical parameters, data were expressed as absolute frequencies (*n*) and percentages (%), with group differences evaluated using Pearson’s chi-square test. When the expected frequencies were less than 5.0, Fisher’s exact test was applied instead.

#### 2.5.1. Vaccine Effectiveness Calculation

Vaccine effectiveness (VE) was calculated using a formula that does not require precise population size data. The screening method is widely used for VE estimation when individual vaccination data are unavailable and has been validated in similar studies of TBE and other vaccine-preventable diseases [9,10]. VE was determined using the observed number of TBE cases in vaccinated (*O_v_*) and unvaccinated (*O_n_*) individuals, as well as the proportions of vaccinated (*r_v_*) and unvaccinated (*r_n_* = 1 − *r_v_*) individuals in the population. The formula applied was as follows:VE=(1−OvrvOnrn×100

This method allows for an accurate estimation of VE without relying on total population numbers. Demographic data, including population size in the Podlaskie Voivodship, were obtained from Central Statistical Office (https://demografia.stat.gov.pl/BazaDemografia, accessed on 23 April 2025). Vaccine uptake in the general population in this region was estimated by aggregating yearly reported numbers of individuals who completed the full TBE vaccine series, obtained from National Institute of Public Health reports (https://wwwold.pzh.gov.pl/oldpage/epimeld/index_p.html, accessed on 23 April 2025). Reporting definitions and diagnostic criteria remained consistent during the 1999–2020 period. For the calculation of VE, we used cases hospitalized in 1999–2020 because the uptake of the TBE vaccine was first reported in 1999. From 1999 to 2020, this totaled 77,882 completed vaccination series, representing around 6.52% of the population in the region. This approach aligns with independent survey findings indicating a coverage rate of 9% [11]. Because individual-level booster data were unavailable, we assumed that long-term protection persisted following primary vaccination and boosters. We therefore estimated coverage using cumulative completed series from 1999 to 2020 and conducted sensitivity analyses to assess how waning immunity might affect VE estimates. We recognize that this cumulative estimate may underrepresent regional or temporal fluctuations in vaccine uptake.

#### 2.5.2. Estimation of the Epidemiological Measures

To characterize trends in TBE seasonality and temporal shifts in disease intensity, we aggregated hospitalizations by month and year-group intervals. Incidence rate ratios (IRRs) were calculated to compare total hospitalizations between defined time intervals (e.g., 1988–1996 vs. 2006–2020) as a measure of long-term trend in disease burden. To assess seasonal changes, we computed peak-to-off-peak ratios and differences using representative high-incidence months (e.g., July) and low-incidence months (e.g., January). Additionally, month-specific odds ratios were derived to quantify relative changes in seasonality between different time periods. Mean hospitalization counts were also calculated for warm (May–October) and cold (November–April) seasons to detect possible shifts in seasonal patterns over time. These metrics aimed to provide a standardized comparison of seasonality across decades and support interpretation of epidemiological changes beyond visual trends.

Analyses were conducted using the R statistical language (version 4.3.3; R Core Team, 2024, Vienna, Austria) on Windows 11 Pro bit 64 (build 22631), using the packages report (version 0.5.8) [12], gtsummary (version 1.7.2) [13], ggplot2 (version 3.5.0) [14], dplyr (version 1.1.4) [15], and tidyr (version 1.3.1) [16].

### 2.6. Ethical Considerations

This study was conducted in accordance with the guidelines for good clinical practice. Ethical approval was obtained from the Bioethical Commission of the Medical University of Bialystok (decision no. APK.002.384.2024).

## 3. Results

### 3.1. Epidemiology

Between 1988 and 2020, a total number of 1518 patients were hospitalized because of TBE. Five hundred eighty-three patients were female (38.4%), and nine hundred thirty-five (61.6%) were male. A total of 13 people were vaccinated, and the remaining 1505 did not receive any TBE vaccines. Among the vaccinated patients, there were two fully vaccinated and protected, nine incompletely vaccinated, and two patients fully vaccinated, not boosted.

The median age of the patients was 43 years, with no significant difference observed between the vaccinated group and the unvaccinated group. The proportion of females and males was nearly identical in both groups. Eight hundred fifty-six patients (56.5%) were inhabitants of cities, and six hundred fifty-nine (43.5%) lived in the countryside, with no differences in the place of residence between the groups.

A comparative analysis across time intervals revealed not only the expected seasonality of TBE but also a trend toward increased hospitalizations during late summer and early autumn in more recent decades (Figure 1). Across the observation periods, the highest midsummer hospitalizations were recorded in July, with 101 admissions in 2006–2020 compared to 82 in 1988–1996. This increase is reflected in an IRR of 1.18 when comparing the most recent interval (2006–2020) to the earliest one (1988–1996), indicating an 18% rise in total hospitalizations. Similarly, the OR for hospitalizations in July versus January between the two extreme time periods was 1.23, highlighting a modest relative increase in peak-season admissions. Seasonal ratios further emphasized this trend, with July-to-January hospitalization ratios rising from 82 in 1988–1996 to 101 in 2006–2020. Warm-season (May–October) hospitalizations consistently surpassed cold-season (November–April) counts, with averages of 65.8 and 7.2 per month, respectively, in 2006–2020 (Table 1). Seasonal differences were also illustrated by a peak-to-off-peak hospitalization difference of 349 and a peak-to-off-peak ratio of 88. The July-to-January hospitalization ratio rose from 82 in 1988–1996 to 101 in 2006–2020, and mean monthly hospitalizations in the warm season increased from 54.8 to 65.8.

### 3.2. Clinical Findings

The characteristics of the study population are presented in Table 2. The duration of the first hospitalization showed significant difference between vaccinated and unvaccinated patients. The median duration of hospitalization for the study group was 19 days. Vaccinated patients had a shorter median hospitalization duration of 14 days (IQR: 12.5–16), compared to 19 days (IQR: 15–27) for unvaccinated patients. Diagnoses were distributed primarily among meningitis (59.7%), encephalomeningitis (31.8%), and meningoencephalomyelitis (8.45%), with no differences in clinical presentation between the groups. Clinical symptoms such as muscle aches, headaches, dizziness, vomiting, and nausea occurred with similar frequency in vaccinated and unvaccinated individuals. The most commonly reported symptom was headache. Meningeal signs such as neck stiffness and Kernig’s and Brudziński’s signs occurred at a similar level in both groups. Furthermore, there was no statistical difference between the two groups of patients when comparing the pyramidal signs. Babinski’s sign, limb paresis, sensory disturbances, and cerebellar symptoms only occurred in unvaccinated patients.

### 3.3. Laboratory Findings

Peripheral blood examination revealed lymphopenia and neutrophilia, with no differences between the vaccinated and unvaccinated patients. CSF cytosis, however, differed between vaccinated and unvaccinated groups. Initially higher in unvaccinated patients (116 vs. 102), cytosis rose sharply in vaccinated individuals by day 1 (138 vs. 106) and day 2 (160 vs. 114). Later, the levels dropped below unvaccinated values at day 12 (34 vs. 65.1) but rose again at days 13, 19, and 23. Similarly, CSF protein levels showed differences between the study groups. Initially higher in unvaccinated patients (64.5 vs. 55.9), the protein levels in vaccinated patients rose on days 1 and 2 after admission, peaked on days 13 (89.4 vs. 61.1) and 19 (138 vs. 59.7), and dropped below the levels in unvaccinated patients by day 23 (33.2 vs. 71.3).

Vaccinated individuals showed a sharp increase in lymphocyte CSF counts early on (day 1: 134 vs. 75.4; day 2: 189 vs. 87.1) (Figure 2). Later, lymphocyte CSF counts dropped below unvaccinated levels (day 12: 29.9 vs. 60.2) and converged by day 19 (41.5 vs. 40.2).

Monocyte counts fluctuated between vaccinated and unvaccinated groups. Early higher counts in vaccinated individuals (17 vs. 15.7) dropped on day 1 (6.9 vs. 14.6) but surged again by day 2 (25.1 vs. 16.1). Later, counts converged, with slight elevations in vaccinated patients by day 19 (3.08 vs. 2.58).

### 3.4. Vaccine Effectiveness

Based on aggregated regional data, cumulative vaccine uptake was estimated at 6.52% for the years 1999–2020 (Appendix A). Vaccine effectiveness was estimated using the screening method, comparing the proportion of vaccinated individuals among TBE cases to the estimated population coverage. The vaccinated denominator comprised all individuals who had completed the 3-dose primary vaccination series between 1999 and 2020, regardless of booster status, as population-level data on booster uptake were unavailable. Accordingly, the numerator included all breakthrough TBE cases (*n* = 4) that occurred in individuals who had completed the primary series—two of whom had received timely boosters and two who had not. This ensured consistency between numerator and denominator definitions. The resulting pooled VE was estimated at 94.4% with a 95% CI of 85.2–97.9%.

To account for the uncertainty in long-term vaccine-induced immunity, we conducted a one-way sensitivity analysis by recalculating VE under the assumption that only 50% or 75% of the 77,882 reported completed vaccination series conferred continued protection over time. For each scenario, adjusted population coverage values were used to compute odds-ratio-based VE estimates, with 95% confidence intervals calculated via the Woolf method [17]. VE ranged from 88.5% (95% CI: 69.3–95.7%) under the 50% protection assumption to 92.5% (95% CI: 79.9–97.2%) under the 75% assumption.

### 3.5. Case Studies of Patients Hospitalized with TBE Despite Vaccination

Due to the small number of vaccinated TBE cases (*n* = 13), no statistical comparisons were made between subgroups (e.g., fully vaccinated vs. incompletely vaccinated). However, we present individual-level clinical symptom profiles descriptively to highlight the heterogeneity of breakthrough TBE cases. Figure 3 summarizes the clinical symptoms in previously vaccinated patients; detailed clinical courses are provided in the Appendix A.

Notably, both patients who were fully vaccinated and protected (patients 3 and 9) developed severe forms of TBE-encephalomeningitis, with prolonged hospitalization (15 and 24 days, respectively). In contrast, patients who were fully vaccinated but not boosted (patients 1 and 7) experienced milder, meningitic forms of the disease, with shorter hospital stays (7 and 14 days) and no encephalitic or myelitic features. Among the incompletely vaccinated patients, disease severity was variable: while most (patients 4, 6, 8, 10, 11, and 12) had moderate or mild courses (meningitis, 12–15 days), some (patients 2, 5, and 13) developed severe disease with encephalitic or myelitic involvement and prolonged hospitalizations (17–26 days).

## 4. Discussion

Tick-borne encephalitis poses a significant public health concern, with the potential for severe neurological complications. Vaccination is the only known way to prevent the disease. Recent evidence indicates that breakthrough TBE in vaccinated individuals is not only possible but may also be associated with a more severe disease course than in unvaccinated patients. Several large-scale epidemiological studies from endemic regions, such as Austria, Germany, and Switzerland, have examined the characteristics and severity of TBE in vaccinated populations [18,19,20,21,22]. Our findings align with these studies in showing that while breakthrough infections are rare, they can present with moderate to severe clinical symptoms. Lotrič-Furlan et al. reported that such cases were characterized by a higher frequency of severe acute illness, increased need for ICU admission, and longer hospital stays, with a notable tendency toward monophasic illness and distinct serological patterns reflecting an anamnestic response [18]. Similarly, Wagner et al. observed that vaccinated patients exhibited more extensive and severe MRI changes, particularly affecting deep brain structures, and consistently experienced significant neurological impairment [22]. Together, these studies suggest that while vaccination remains highly effective in preventing TBE, breakthrough infections—particularly in older adults or those with waning immunity—can result in unexpectedly severe outcomes. Our long-term, retrospective study of TBE cases in a highly endemic region confirms that breakthrough infections can occur despite adherence to recommended schedules. Although the number of vaccinated individuals was small (*n* = 13), we present descriptive comparisons across vaccination subgroups to illustrate the diversity of clinical presentations. Due to sample size limitations, these findings provide exploratory insight into breakthrough cases. Such case-level analyses are valuable given the rarity of TBE despite vaccination and the lack of large-scale data on breakthrough infections in this setting. Notably, severe disease—including encephalomeningitis with prolonged hospitalization—was observed in some patients who were fully vaccinated and considered protected. In contrast, patients who were fully vaccinated but not boosted experienced the mildest clinical courses, presenting with meningitis only and requiring shorter hospital stays. Among the incompletely vaccinated individuals, disease severity was heterogeneous: while most had moderate courses, a subset developed severe neurological involvement. These findings suggest that vaccination, while highly effective at reducing the risk of TBE, does not guarantee protection from severe disease in all cases.

Laboratory findings in vaccinated patients showed dynamic CSF changes, with early increases in cytosis and lymphocytes, followed by declines below the unvaccinated levels, indicating a strong and well-regulated immune response. Neurological symptoms, including meningeal signs and consciousness disturbances, were less frequent in vaccinated individuals, hinting at partial protection. Blood parameters revealed lower lymphocyte counts at admission with transient surges later, while other markers showed no significant differences. It is important to pay attention to the role of lymphocytes in the course of the disease. There are studies suggesting that lymphocytes play a key role in alleviating the course of the viral disease. For example, research on COVID-19 showed that patients with a higher number of lymphocytes had a milder course of the disease. In turn, lymphopenia was associated with a more severe course and a worse prognosis. One study found that patients with severe COVID-19 had significantly lower numbers of CD3, CD4, and CD8 lymphocytes compared to patients with milder cases [23].

Breakthrough tick-borne encephalitis (TBE) in fully vaccinated individuals is rare but demonstrates that factors beyond vaccination—such as host immunity, comorbidities, age, or viral characteristics—can influence clinical severity. Several large series from Germany found no evidence that vaccine-breakthrough infections (VBI) are clinically more severe than primary (vaccine-naïve) infections [19]. Likewise, Swiss data showed no increase in the VBI risk when booster intervals were prolonged, and population acceptance of TBE immunization has remained high [24]. Both reports place the VBI rate between 4% and 5.8%. Emerging immunological data are more nuanced. A recent study observed that VBI cases mounted weaker antibody responses to the NS1 protein but retained robust T-cell reactivity; these cases were frequently of moderate-to-severe clinical intensity. In contrast, higher NS1 and EDIII antibody titers in unvaccinated patients correlated with milder disease, suggesting that VBI may alter the humoral profile and, in turn, affect the outcome [20]. Misdiagnosis is an additional concern. Grgič-Vitek et al. highlighted that cross-reactivity with other flaviviruses and atypical serodynamics can lead to false attribution of TBE in vaccinated persons [21]. Overall, breakthrough infections remain uncommon, and the field effectiveness of the licensed vaccines is consistently high—about 96% after ≥3 doses [5]. In the present series, only four hospitalized cases occurred in people who had completed the primary schedule over a 22-year period, underscoring the robustness of protection. To our knowledge, this is the first estimate of TBE vaccine effectiveness derived from a highly endemic region in Poland. Comparable VE studies have been published for Austria, Germany, Latvia, Switzerland, and Sweden, but none previously for Poland despite its long-recognized endemicity.

Our long-term hospital surveillance also quantifies subtle shifts in seasonality. While the classic summer peak persists, we observed a gradual rise in late-summer and early-autumn admissions, a pattern that may reflect climatic and ecological changes influencing tick activity or changing human exposure patterns. Similar extensions of the transmission season have been reported elsewhere in Central Europe [25].

Our study has several limitations. First, the estimated vaccination coverage may be somewhat underestimated. The annual reported uptake of the TBE vaccine was consistently low (around 0.3%), which appears to underrepresent the true coverage. A prior questionnaire-based study conducted in the same region suggested higher uptake—approximately 9% [11]. Additionally, TBE vaccination has been available in Poland since the 1970s, meaning that long-term retention of vaccinated individuals may result in cumulative coverage reaching 5–6%, even with low annual uptake. While the exact historical coverage remains uncertain, we believe that our aggregated estimate of 6.5% is a reasonable and conservative approximation.

Second, the VE estimate relies on assumptions regarding the duration of vaccine-induced immunity and population-level coverage. These assumptions do not fully account for waning immunity or gaps in booster adherence. Nonetheless, we aligned our numerator and denominator definitions to include all individuals who had completed the primary vaccination series, regardless of booster status, and incorporated sensitivity analyses to explore these uncertainties. Although based on a small number of vaccinated cases (*n* = 4), the VE estimate is internally consistent and comparable to other published estimates. Notably, the 2015 Ebola ring vaccination trial (Henao-Restrepo et al. [26]) reported vaccine effectiveness based on zero breakthrough cases in the vaccinated group. Our VE analysis similarly includes formal sensitivity testing and wide confidence intervals, offering valuable real-world evidence of vaccine performance in a highly endemic region.

Third, the study is limited by reliance on single-center hospital data, which captures approximately 50% of all TBE cases reported in the region. This may result in the underestimation of true regional incidence. In addition, using average population figures across a 21-year period for VE calculation may obscure temporal shifts in population size or demographics. Finally, because our dataset includes only hospitalized patients and excludes mild or asymptomatic cases, we cannot draw definitive conclusions about the association between the vaccination status and disease severity. While some trends were observed, these remain exploratory and should be interpreted cautiously.

Despite these limitations, our results provide meaningful insights. The use of laboratory-confirmed cases with well-documented vaccination histories offers a reliable basis for assessing vaccine performance in real-world conditions.

## 5. Conclusions

Our 33-year hospital-based study provides the most detailed description to date of breakthrough TBE in Poland. Breakthrough cases were exceptionally rare (13/1 518), and when they occurred, their clinical presentation was broadly comparable with that of unvaccinated patients, although cerebrospinal fluid findings suggested a more rapid immune response in vaccinees. Long-term surveillance also documents subtle but important epidemiological shifts: a steady rise in late-summer and early-autumn admissions and a modest overall increase in annual hospitalizations. These patterns are consistent with reports that climate- and ecology-driven changes in tick activity are lengthening the transmission season in Central Europe. Persistently low vaccine uptake in the endemic northeast, therefore, represents a missed prevention opportunity; expanded access, clearer booster guidance, and integration of vaccination records into routine surveillance are immediate public health priorities.

Finally, using a denominator of all residents who completed the three-dose primary schedule and a numerator of the four corresponding breakthrough cases, we estimated a vaccine effectiveness of 94% (95% CI 85–98%). Although this figure aligns with field data from neighboring countries, it is based on few events and should be interpreted with caution until larger datasets become available.

## Figures and Tables

**Figure 1 vaccines-13-00665-f001:**
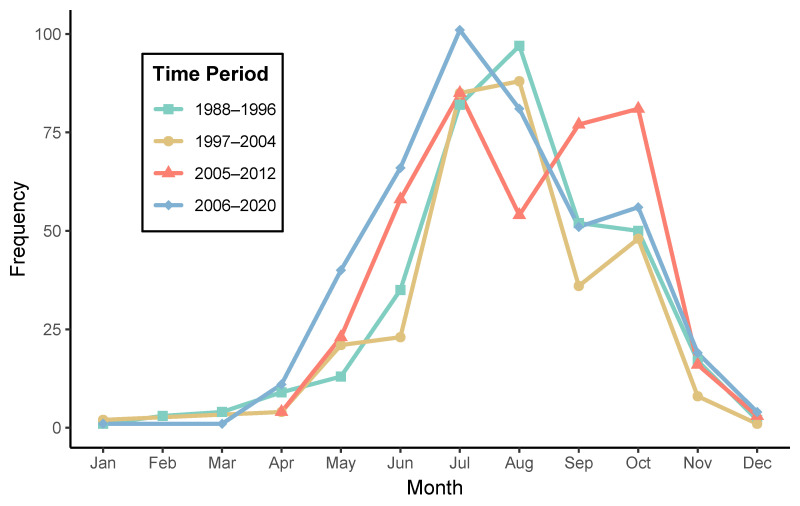
Annual frequency of hospitalizations of patients with TBE, stratified by four observation time periods.

**Figure 2 vaccines-13-00665-f002:**
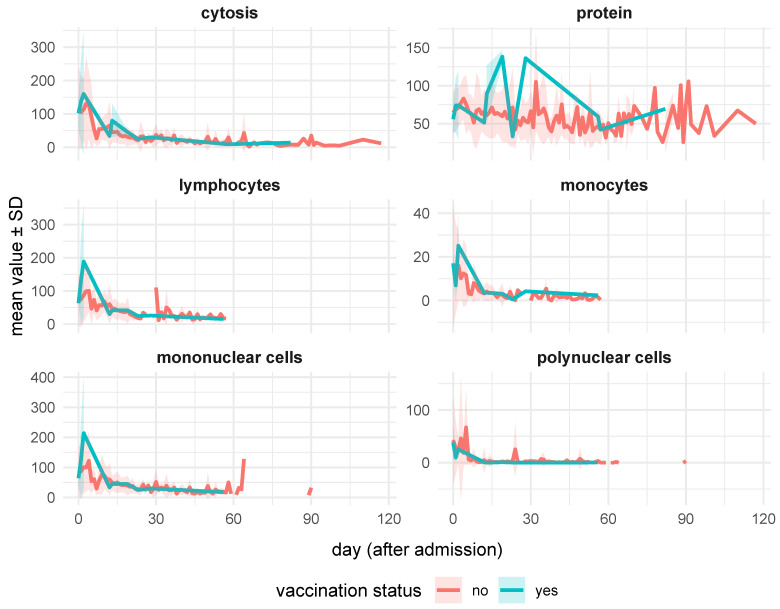
Concentration of the CSF parameters in TBE patients upon admission stratified by vaccination status.

**Figure 3 vaccines-13-00665-f003:**
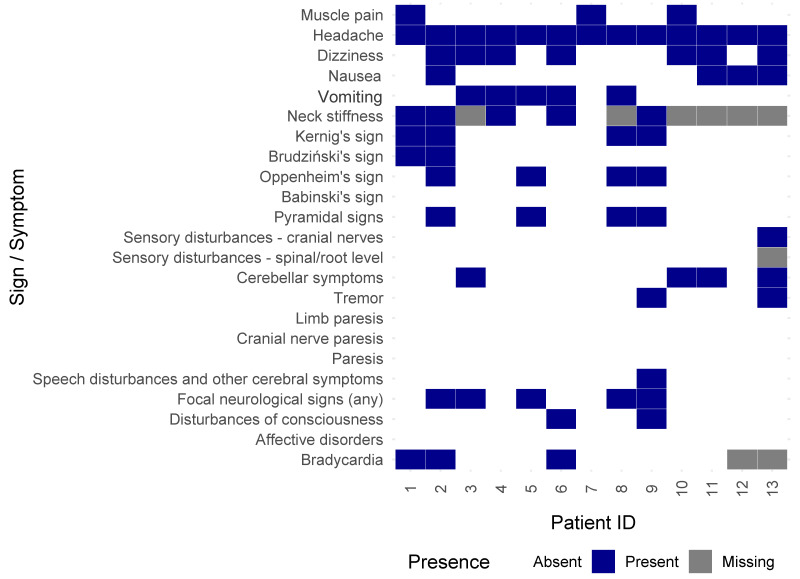
Symptom distribution across 13 vaccinated TBE patients. These data are presented for descriptive purposes only. No statistical comparisons between vaccinated subgroups were performed.

**Table 1 vaccines-13-00665-t001:** Seasonal distribution of TBE hospitalizations across different time intervals. The number of months included in each seasonal segment may vary (e.g., 3–5 months for some winter intervals) due to the absence of reported hospitalizations during certain winter months, particularly January to March. This reflects the near-zero incidence of TBE in colder months, as shown in Figure 1.

Time Period	Season	Total Hospitalizations	Number of Months	Average Count
1988–1996	Cold	36	6	6.0
	Warm	329	6	54.8
1997–2004	Cold	15	4	3.8
	Warm	301	6	50.2
2005–2012	Cold	23	3	7.7
	Warm	378	6	63.0
2006–2020	Cold	36	5	7.2
	Warm	395	6	65.8

**Table 2 vaccines-13-00665-t002:** Characteristics of the studied parameters for overall sample along with stratification by the vaccination status against TBE for patients hospitalized with TBE.

Characteristic	*n*	Overall Sample	Vaccination Against TBE		*p* ^c^
			Yes *n*_1_ = 13 ^a^	No *n*_2_ = 1505 ^a^	
Demographic and general characteristics					
Age, years	1516	43.00 (31.00, 57.00) ^b^	42.00 (38.00, 47.00) ^b^	44.00 (31.00, 57.00) ^b^	0.743 ^e^
Sex:	1518				0.577 ^d^
female		583 (38.41%)	6 (46.15%)	577 (38.34%)	
male		935 (61.59%)	7 (53.85%)	928 (61.66%)	
Locality:	1515				0.648 ^d^
city		856 (56.50%)	6 (50%)	850 (56.55%)	
village		659 (43.50%)	6 (50%)	653 (43.45%)	
Periods of hospitalization observations, seasonality, and durations					
Periods of observation:	1518				0.051
1988–1996		370 (24.37%)	0 (0%)	370 (24.58%)	
1997–2004		316 (20.82%)	2 (15.38%)	314 (20.86%)	
2005–2012		401 (26.42%)	7 (53.85%)	394 (26.18%)	
2013–2020		431 (28.39%)	4 (30.77%)	427 (28.37%)	
Season of hospitalization:	1514				0.541
winter		17 (1.12%)	0 (0%)	17 (1.13%)	
spring		130 (8.59%)	2 (15.38%)	128 (8.53%)	
summer		856 (56.54%)	8 (61.54%)	848 (56.50%)	
autumn		511 (33.75%)	3 (23.08%)	508 (33.84%)	
Duration of first hospitalization, days	1449	19.00 (15.00, 27.00) ^b^	14.00 (12.50, 16.00) ^b^	19.00 (15.00, 27.00) ^b^	0.015 ^e^
General symptoms					
Diagnosis:	1480				1.000
encephalomeningits		471 (31.82%)	4 (33.33%)	467 (31.81%)	
meningitis		884 (59.73%)	7 (58.33%)	877 (59.74%)	
meningoencephalomyelitis		125 (8.45%)	1 (8.33%)	124 (8.45%)	
Muscle pain	1309	332 (25.36%)	3 (23.08%)	329 (25.39%)	1.000
Headache	1315	1252 (95.21%)	13 (100%)	1239 (95.16%)	1.000 ^d^
Dizziness	1314	538 (40.94%)	7 (53.85%)	531 (40.81%)	0.342 ^d^
Nausea	1315	307 (23.35%)	4 (30.77%)	303 (23.27%)	0.515
Vomiting	1315	441 (33.54%)	5 (38.46%)	436 (33.49%)	0.770 ^d^
Other symptoms	1320	300 (22.73%)	4 (30.77%)	296 (22.65%)	0.507
Neurological and other symptoms					
Neck stiffness	802	576 (71.82%)	5 (71.43%)	571 (71.82%)	1.000 ^d^
Kernig’s sign	1307	652 (49.89%)	4 (30.77%)	648 (50.08%)	0.166
Brudziński’s sign	1309	344 (26.28%)	2 (15.38%)	342 (26.39%)	0.532
Oppenheim’s sign	1369	448 (32.72%)	4 (30.77%)	444 (32.74%)	1.000
Babinski’s sign	1371	121 (8.82%)	0 (0%)	121 (8.90%)	0.620
Pyramidal signs	1375	480 (34.91%)	4 (30.77%)	476 (34.95%)	1.000
Sensory disturbances—cranial nerves	1373	36 (2.62%)	1 (7.69%)	35 (2.57%)	0.293
Sensory disturbances—spinal/root level	1371	64 (4.67%)	0 (0%)	64 (4.71%)	1.000
Cerebellar symptoms	1376	263 (19.11%)	4 (30.77%)	259 (19%)	0.288
Tremor	1376	133 (9.67%)	2 (15.38%)	131 (9.61%)	0.363
Limb paresis	1380	114 (8.26%)	0 (0%)	114 (8.34%)	0.617
Cranial nerve paresis	1381	56 (4.06%)	0 (0%)	56 (4.09%)	1.000
Speech disturbances and other cerebral symptoms	1383	26 (1.88%)	1 (7.69%)	25 (1.82%)	0.219
Focal neurological signs (any)	1382	646 (46.74%)	5 (38.46%)	641 (46.82%)	0.548 ^d^
Disturbances of consciousness	1386	393 (28.35%)	2 (15.38%)	391 (28.48%)	0.372
Bradycardia	1327	130 (9.80%)	3 (27.27%)	127 (9.65%)	0.084
Differential blood count at admission					
Lymphocytes	1062	1515.85 (1150.00, 2008.89)	1167.26 (664.78, 1426.76)	1520.00 (1152.82, 2010.67)	0.087
Neutrophils	747	6751.80 (4862.50, 8845.00)	7310.00 (5930.00, 9070.00)	6745.56 (4856.25, 8839.85)	0.999
Monocytes	858	700.00 (158.50, 1000.00)	800.00 (420.48, 880.00)	700.00 (130.00, 1000.00)	0.716
Eosinophils	803	10.00 (0.00, 76.09)	0.00 (0.00, 19.01)	10.00 (0.00, 77.00)	0.317
Basophils	803	0.00 (0.00, 20.00)	8.92 (0.00, 9.85)	0.00 (0.00, 20.00)	0.651

^a^ n (%); ^b^ Median (IQR); ^c^ Fisher’s exact test; ^d^ Pearson’s chi-squared test; ^e^ Wilcoxon rank sum test.

## Data Availability

The data presented in this study are available on request from the corresponding author.

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
