# Peer review of "Breakthrough Tick-Borne Encephalitis and Epidemiological Trends in an Endemic Region in Poland: A Retrospective Hospital-Based Study, 1988–2020"

_vaccines, 2025, doi:10.3390/vaccines13070665_

Round 1
Reviewer 1 Report
Comments and Suggestions for Authors
This is a useful study which provides, for the first time, an estimate of TBE vaccine effectiveness in Poland - a country with regions with high TBE incidence.
I have four major concerns for the paper:
- When deriving an estimate of vaccine effectiveness using the screening method, it is essential to provide sufficient details in the methods section as to how vaccine uptake of the general population (or survey of the general population) is derived. Unfortunately, while the manuscript provides a link to a government surveillance report that reports the number of TBE vaccine doses administered each year, the manuscript does not explain how the population vaccine uptake is derived from this government surveillance report. Furthermore, the estimated vaccine uptake of the general population, which the manuscript suggests is 0.3% is much too low. Surveys of the general population in Poland (please see Pilz et al. Ticks Tickborne Dis 2022) indicate that 12% of the general population in Poland has received at least 3 doses of a TBE vaccine (on schedule with the vaccination schedule). estimate is derived.
- The authors need to make clear that the definition of TBE is a TBE virus infection with signs/symptoms of CNS involvement. And that the TBE cases in this study are all TBE cases (i.e., no non-CNS TBEV-infected cases) and no non-hospitalized cases. This is important because this means that mild TBEV-infected cases are not included in the study, which has implications for trying to assess the association between disease severity and vaccination.
- The manuscript provides too much emphasis on postulated association between TBE vaccination and more severe TBE; this postulated association has been shown not to be present in study by Nygren (reference #5) and others; the observed association in some studies between TBE vaccination and more severe TBE is due to those studies not including mild TBE cases in their study. Unfortunately, this is the same problem in this study in Poland; there are no non-hospitalized TBEV-infected cases and no non-CNS TBEV-infected cases. Therefore, the authors should reduce the focus on the disease severity and vaccination in this study.
- I don't find it helpful to compare the 13 TBE cases who received one or more TBE vaccine dose to each other (as is done with Table 2). There are so few cases: deciding that the 2 TBE cases that are fully vaccinated and protected are different from the 2 TBE cases that are fully vaccinated but without a booster does not make sense. The numbers are too small.
Other comments
- Title: There are three problems with the title: (1) I would drop "Breakthrough" because it should not be the major focus of the study, (2) I would drop "outcomes" because the information on outcomes is limited, and (3) it is not really "retrospective cohort" study because the patients are not followed from exposure to outcome. A better title is "A hospital-based study of tick-borne encephalitis vaccine effectiveness in a highly endemic region of Poland, 1988-2020"
- Abstract (background) at lines 13-14: change "...compare the clinical course and estimate vaccine effectiveness (VE) for TBE among vaccinated and unvaccinated..." to ",,,describe the clinical course of vaccinated and unvaccinated TBE cases and estimate TBE vaccine effectiveness (VE)...."
- Abstract (results) at lines 25-26: drop "while fully vaccinated but not boosted patients experienced milder disease". There are only two fully vaccinated but not boosted patients. There are insufficient numbers to evaluate if the clinical course is different from these two versus other vaccinated TBE cases.
- Introduction section at lines 38: It is not correct to say "TBEV is endemic" (since viruses are not endemic). Change to "TBEV infection is endemic" or, alternatively, "TBE is endemic" (since diseases and infections are endemic).
- Introdution section at line 38: Authors need to define "TBE" as a TBEV infection that results in signs and/or symptoms of CNS inflammation. Then can say "TBE is endemic" (once TBE has been defined).
- Introduction section at lines 48-49: need to clarlify that there are two TBE vaccines available in Europe (or, alternatively, available in Poland). There are several other TBE vaccines available in Russia and China (but only two TBE vaccines in Europe).
- Introduction section at lines 65-66: change "overall effectiveness of TBE vaccines in preventing severe disease." to "overall effectiveness of TBE vaccines."
- Materials and Methods section at line 69: change "verified" to "laboratory confirmed"
- Materials and Methods section at line 73: change "verified" to "confirmed".
- Materials and Methods section at line 85: is confusing because it is stated that "Patients were categorized into two groups depending on vaccine status", but then at lines 93-98, the vaccinated are further classified into three groups. At line 85, would be better drop the following sentance at line 85: "Patients were categorized into two groups depending on vaccine status" (the classification of patients by vaccine status is clarified later, so not needed here).
- Materials and Methods at line 99: you use the term "fully vaccinated not protected" but later in the manuscript, you use the term "fully vaccinated but not boostered" (see line 163). Be consistent and use just one term.
- Matrials and Methods at lines 118-119: it is good that the link is provided to the government report on vaccination, but it is not possible to determine (from the information at the link) how the vaccine uptake (or the proportion of the general population that is fully-vaccined and protected) is derived from the information on number of doses of vaccine administered (which is what appears to be provided at the link). The credibility of using the screening method to estimate VE (i.e., to enable the readers to judge the extent of bias, particularly selection bias, in the screening method) depends on providing a clear explanation of how the estimate of vaccine uptake among the general population is derived. I think an additional paragraph is needed in the Materials and Methods section which provides an explanation of how vaccine uptake of the general population is derived.
- Materials and Methods at lines 130-139: I suggest droping the calculation of peak-to-off-peak; this is so obvious and is easily seen in the figure. It is not necessary to create a statistical test to show that there is seasonal variation (which is well known).
- Results section at line 180-182: I would drop the peak-to-off-peak calculations.
- Results section at lines 182-183: drop "These finding suggest a prolonged and intensified TBE season in recent decades". This is an interpretation of the data. Interpretations go in the discussion section not in the results section.
- Results section at line 184-185: drop "Epidemiology of the disease is changing due to a warming climate". This is not a study result, this is a judgement. Judgements go in the discussion section not in the results section.
- Results section at lines 185-187: drop the statement that "This section may be divided by subheadings." No directions (or methods) in the results section (just results).
- Table 1: it is not clear why the number of months in each time segment is not 6 months; how can there be a 3 month, 4 month, and 5 month segment?
- Results section at lines 209-210: the finding of slightly more bradycardia in TBE cases that are vaccinated than unvaccinated TBE cases is not worth highlighting. The p value is 0.08. This observation can be explained by very small numbers of vaccinated TBE cases.
- Results section at line 224: there is no Figure 4. (and why is Figure 4 mentioned before Figure 2?).
- Results section at lines 227-228: Drop "These patterns suggest early immune activation and recurring late-phase responses in vaccinated patients." This is not a result, but an interpretation. Move interpretations to the discussion section.
- Results section at line 236: Drop "This suggests an early immune boost folled by normalization". This is not a result but an interpretation.
- Results section at line line 240: Drop "suggesting potential immune benefits". This is an interpretation not a result".
- Results section at line 247: stating the only the data from 1999-2020 was used in the VE estimation belongs in the methods section not the results section.
- Results section at line 254-255: Drop "These results indicate a high level........in other studies". This is an judgement, which should be moved to the discussion section.
- Results section: Drop section 3.5 (lines 258-268) and drop Figure 3. I don't see how there can be any interpretation about the differences in 13 TBE cases in which two are fully-vaccinated and protected, two are fully-vaccinated but not boosted, and nine are incompletely vaccinated. These numbers are too small to compare with each other. No way can conclude two patients are different from two other patients.
- Discussion at lines 279-288: I would drop these lines. A number of studies have demonstrated that the potential association between TBE vaccination and increased disease severity is likely fallecous due to the studies not including a sufficent number of TBEV-infected patients with mild disease (i.e., not hospitalized or without CNS inflammation). If studies only include TBE cases (like this study), then the studies only include the severe TBE cases, so trying to show an association between disease severity and vaccination is difficult (because all the TBE cases are severe).
- Discussion section at line 304: I would not highlight the bradycardia.
- Discussion section at lines 348-355: I would drop the paragraph on antibody-dependent enhancement. This has only been demonstrated with dengue and has consistently been shown to not be present with TBEV infection. There is no need to bring this up in this study (since this study has no information on ADE).
- Discussion section at lines 378-382. I diagree with the concluding focus of this paper. The most notable and important findings of this study is that, for the first time, TBE VE estimates are available from Poland. And the TBE VE is high. So the public health message should be emphasized: TBE is a potentially serious and life threatening disease that can be prevented by vaccination, and TBE is endemic in Poland.....but only a small proportion of the general population in Poland are vaccinated. So there need to be enhanced efforts to increase TBE vaccine uptake in Poland.
Author Response
This is a useful study which provides, for the first time, an estimate of TBE vaccine effectiveness in Poland - a country with regions with high TBE incidence.
Dear Reviewer. Thank you for your detailed and insightful review. Our paper was improved and all points that you have raised were addressed.
- When deriving an estimate of vaccine effectiveness using the screening method, it is essential to provide sufficient details in the methods section as to how vaccine uptake of the general population (or survey of the general population) is derived. Unfortunately, while the manuscript provides a link to a government surveillance report that reports the number of TBE vaccine doses administered each year, the manuscript does not explain how the population vaccine uptake is derived from this government surveillance report. Furthermore, the estimated vaccine uptake of the general population, which the manuscript suggests is 0.3% is much too low. Surveys of the general population in Poland (please see Pilz et al. Ticks Tickborne Dis 2022) indicate that 12% of the general population in Poland has received at least 3 doses of a TBE vaccine (on schedule with the vaccination schedule).
Thank you for pointing this out. Estimated vaccine coverage of 0.3% mentioned in the text refers to the annual reported number of vaccinated inhabitants. We believe that over the years, the aggregated number of vaccinees averages around 10%. In our previous survey study from the same region this percentage was 9%. We used 6.49% as the aggregated number of vaccinated individuals calculated from the number of official reports from 19 years. This might be underestimated, but since there are no reliable data on the total vaccine coverage we have to use this method to calculate VE. This is now explained in the text in lines paragraph 2.5.1.
2. The authors need to make clear that the definition of TBE is a TBE virus infection with signs/symptoms of CNS involvement. And that the TBE cases in this study are all TBE cases (i.e., no non-CNS TBEV-infected cases) and no non-hospitalized cases. This is important because this means that mild TBEV-infected cases are not included in the study, which has implications for trying to assess the association between disease severity and vaccination.
The definition for TBE is now included in paragraph 2.2
3. The manuscript provides too much emphasis on postulated association between TBE vaccination and more severe TBE; this postulated association has been shown not to be present in study by Nygren (reference #5) and others; the observed association in some studies between TBE vaccination and more severe TBE is due to those studies not including mild TBE cases in their study. Unfortunately, this is the same problem in this study in Poland; there are no non-hospitalized TBEV-infected cases and no non-CNS TBEV-infected cases. Therefore, the authors should reduce the focus on the disease severity and vaccination in this study.
Thank you for this comment. We agree with the reviewer that due to the absence of non-hospitalized and non-CNS TBE cases in our study population, it is not possible to draw conclusions about the association between vaccination and disease severity. We have revised the Discussion to clarify this limitation and to ensure the tone does not suggest a causal link. The updated text now emphasizes that our findings are exploratory and should be interpreted with caution, in line with the reviewer’s guidance.
4. I don't find it helpful to compare the 13 TBE cases who received one or more TBE vaccine dose to each other (as is done with Table 2). There are so few cases: deciding that the 2 TBE cases that are fully vaccinated and protected are different from the 2 TBE cases that are fully vaccinated but without a booster does not make sense. The numbers are too small.
We appreciate this comment and fully agree that statistical comparisons between such small subgroups are not appropriate. However, we believe that presenting descriptive comparisons of clinical and demographic features in vaccinated individuals remains valuable, especially given the rarity of breakthrough cases. We have revised the text to clearly state that these subgroup analyses are exploratory in nature and are not intended to support definitive conclusions. This framing helps clarify the descriptive intent of Table 2 and ensures readers interpret the data appropriately.
Other comments
- Title: There are three problems with the title: (1) I would drop "Breakthrough" because it should not be the major focus of the study, (2) I would drop "outcomes" because the information on outcomes is limited, and (3) it is not really "retrospective cohort" study because the patients are not followed from exposure to outcome. A better title is "A hospital-based study of tick-borne encephalitis vaccine effectiveness in a highly endemic region of Poland, 1988-2020"
The title was modified.
2. Abstract (background) at lines 13-14: change "...compare the clinical course and estimate vaccine effectiveness (VE) for TBE among vaccinated and unvaccinated..." to ",,,describe the clinical course of vaccinated and unvaccinated TBE cases and estimate TBE vaccine effectiveness (VE)...."
Corrected
3. Abstract (results) at lines 25-26: drop "while fully vaccinated but not boosted patients experienced milder disease". There are only two fully vaccinated but not boosted patients. There are insufficient numbers to evaluate if the clinical course is different from these two versus other vaccinated TBE cases.
Corrected
4. Introduction section at lines 38: It is not correct to say "TBEV is endemic" (since viruses are not endemic). Change to "TBEV infection is endemic" or, alternatively, "TBE is endemic" (since diseases and infections are endemic).
Corrected
5. Introdution section at line 38: Authors need to define "TBE" as a TBEV infection that results in signs and/or symptoms of CNS inflammation. Then can say "TBE is endemic" (once TBE has been defined).
This is explained in the first sentence.
6. Introduction section at lines 48-49: need to clarlify that there are two TBE vaccines available in Europe (or, alternatively, available in Poland). There are several other TBE vaccines available in Russia and China (but only two TBE vaccines in Europe).
Thank you for pointing this out. We added “in Europe”
7. Introduction section at lines 65-66: change "overall effectiveness of TBE vaccines in preventing severe disease." to "overall effectiveness of TBE vaccines."
Corrected
8. Materials and Methods section at line 69: change "verified" to "laboratory confirmed"
Corrected
9. Materials and Methods section at line 73: change "verified" to "confirmed".
Corrected
10. Materials and Methods section at line 85: is confusing because it is stated that "Patients were categorized into two groups depending on vaccine status", but then at lines 93-98, the vaccinated are further classified into three groups. At line 85, would be better drop the following sentance at line 85: "Patients were categorized into two groups depending on vaccine status" (the classification of patients by vaccine status is clarified later, so not needed here).
We agree, this sentence was removed.
11. Materials and Methods at line 99: you use the term "fully vaccinated not protected" but later in the manuscript, you use the term "fully vaccinated but not boostered" (see line 163). Be consistent and use just one term.
Thank you for catching this. We decided on “fully vaccinated, not boosted”
12. Matrials and Methods at lines 118-119: it is good that the link is provided to the government report on vaccination, but it is not possible to determine (from the information at the link) how the vaccine uptake (or the proportion of the general population that is fully-vaccined and protected) is derived from the information on number of doses of vaccine administered (which is what appears to be provided at the link). The credibility of using the screening method to estimate VE (i.e., to enable the readers to judge the extent of bias, particularly selection bias, in the screening method) depends on providing a clear explanation of how the estimate of vaccine uptake among the general population is derived. I think an additional paragraph is needed in the Materials and Methods section which provides an explanation of how vaccine uptake of the general population is derived.
We agree, the method to calculate vaccine coverage was explained further.
13. Materials and Methods at lines 130-139: I suggest droping the calculation of peak-to-off-peak; this is so obvious and is easily seen in the figure. It is not necessary to create a statistical test to show that there is seasonal variation (which is well known).
We appreciate this suggestion. While seasonality of TBE is indeed well recognized, our aim was not simply to confirm its presence, but to quantify the magnitude and shifts in seasonal patterns over time. The use of peak-to-off-peak ratios, odds ratios, and incidence rate ratios allowed us to systematically compare temporal trends and illustrate potential epidemiological changes, such as the intensification or prolongation of the transmission season. We have clarified this purpose in the revised text to better reflect the added value of this analysis.
14. Results section at line 180-182: I would drop the peak-to-off-peak calculations.
Same as above
15. Results section at lines 182-183: drop "These finding suggest a prolonged and intensified TBE season in recent decades". This is an interpretation of the data. Interpretations go in the discussion section not in the results section.
Corrected
16. Results section at line 184-185: drop "Epidemiology of the disease is changing due to a warming climate". This is not a study result, this is a judgement. Judgements go in the discussion section not in the results section.
We agree. Corrected
17. Results section at lines 185-187: drop the statement that "This section may be divided by subheadings." No directions (or methods) in the results section (just results).
Thank you for noticing. Corrected.
18. Table 1: it is not clear why the number of months in each time segment is not 6 months; how can there be a 3 month, 4 month, and 5 month segment?
Thank you for this observation. The reason for the variation in the number of months within winter segments (e.g., 3–5 months rather than 6) is that TBE hospitalizations did not occur uniformly throughout all winter months. As shown in Figure 1, hospitalization counts from January through March are consistently near zero, and in some years, there were no hospitalizations during these months. As such, the effective duration of the "off-peak" season varies slightly depending on actual case occurrence, which is reflected in the data presented in Table 1.
19. Results section at lines 209-210: the finding of slightly more bradycardia in TBE cases that are vaccinated than unvaccinated TBE cases is not worth highlighting. The p value is 0.08. This observation can be explained by very small numbers of vaccinated TBE cases.
Agree. Removed.
20. Results section at line 224: there is no Figure 4. (and why is Figure 4 mentioned before Figure 2?).
Thank you for noticing. Figure 4 was present in a previous version of the paper, this sentence should have been removed as well.
21. Results section at lines 227-228: Drop "These patterns suggest early immune activation and recurring late-phase responses in vaccinated patients." This is not a result, but an interpretation. Move interpretations to the discussion section.
Removed
22. Results section at line 236: Drop "This suggests an early immune boost folled by normalization". This is not a result but an interpretation.
Removed
23. Results section at line line 240: Drop "suggesting potential immune benefits". This is an interpretation not a result".
Removed
24. Results section at line 247: stating the only the data from 1999-2020 was used in the VE estimation belongs in the methods section not the results section.
This sentence was moved to the methods section
25. Results section at line 254-255: Drop "These results indicate a high level........in other studies". This is an judgement, which should be moved to the discussion section.
Sentence removed.
26. Results section: Drop section 3.5 (lines 258-268) and drop Figure 3. I don't see how there can be any interpretation about the differences in 13 TBE cases in which two are fully-vaccinated and protected, two are fully-vaccinated but not boosted, and nine are incompletely vaccinated. These numbers are too small to compare with each other. No way can conclude two patients are different from two other patients.
We appreciate this comment and fully agree that statistical comparisons between vaccinated subgroups (e.g., fully vaccinated vs. incompletely vaccinated) are not appropriate due to the small sample size. Accordingly, our statistical analysis (Table 2) compares only the overall vaccinated group (n=13) to the unvaccinated cohort. Section 3.5 and Figure 3 are included solely to provide a descriptive overview of individual symptom patterns among breakthrough cases. Given the rarity of such cases and the lack of large datasets, we believe this type of case-level visualization adds value for clinicians and researchers. We have revised the text and figure legend to clarify this descriptive intent and avoid misinterpretation.
27. Discussion at lines 279-288: I would drop these lines. A number of studies have demonstrated that the potential association between TBE vaccination and increased disease severity is likely fallecous due to the studies not including a sufficent number of TBEV-infected patients with mild disease (i.e., not hospitalized or without CNS inflammation). If studies only include TBE cases (like this study), then the studies only include the severe TBE cases, so trying to show an association between disease severity and vaccination is difficult (because all the TBE cases are severe).
We agree with the reviewer. As mentioned above, the updated text now emphasizes that our findings are exploratory and should be interpreted with caution, in line with the reviewer’s guidance.
28. Discussion section at line 304: I would not highlight the bradycardia.
We agree. Any mention of bradycardia was removed from the text.
29. Discussion section at lines 348-355: I would drop the paragraph on antibody-dependent enhancement. This has only been demonstrated with dengue and has consistently been shown to not be present with TBEV infection. There is no need to bring this up in this study (since this study has no information on ADE).
True. We removed this paragraph.
30. Discussion section at lines 378-382. I diagree with the concluding focus of this paper. The most notable and important findings of this study is that, for the first time, TBE VE estimates are available from Poland. And the TBE VE is high. So the public health message should be emphasized: TBE is a potentially serious and life threatening disease that can be prevented by vaccination, and TBE is endemic in Poland.....but only a small proportion of the general population in Poland are vaccinated. So there need to be enhanced efforts to increase TBE vaccine uptake in Poland.
We appreciate this suggestion. The conclusion section has been rewritten to strengthen the main findings of this study
Reviewer 2 Report
Comments and Suggestions for Authors
The manucript by Sulik-Wakulinska et al analyses clinical characteristics and routine blood counts in 1,518 TBE patients hospitalized at University Clinic in Bialystok from 1988-2020, including 13 patients with a history of previous TBE vaccination. The authors found that breakthrough cases are very rare, and occurred in only 0.9% of cases. The estimated vaccine effectiveness (from 1999-2020) was 97.2%. Vaccinated patients had a shorter hospital stay compared to unvaccinated, and showed milder disease, apparently depending on whether they were fully vaccinated and received booster vaccination.
The large cohort provides valuable descriptive data on the clinical characteristics of TBE patients, but there are major problems with regard to vaccine effectiveness estimates and interpretation of findings observed in vaccinated patients.
Vaccine effectiveness estimates rely on the proportion of the population that is vaccinated. One major issue that it is not clear what this vaccinated (completely, incompletely) proportions were in the population investigated during the study period. Without clear, region- and time-specific vaccination coverage data, vaccine effectiveness could be substantially over- or under-estimated. The authors should clearly indicate these data.
Another major issue is that only 13 vaccinated (“breakthrough”) cases are included. While this small number can be expected in a population with low vaccination coverage, the stratified analysis of disease severity is underpowered. The subgroup is too small and diverse to draw conclusions about severity and outcomes.
The Discussion should include several large epidemiological studies from Austria, Germany, and other endemic regions that have addressed TBE severity in vaccinated versus non-vaccinated patients.
Minor
The text in lines 185-187 was probably included by mistake.
Author Response
The manucript by Sulik-Wakulinska et al analyses clinical characteristics and routine blood counts in 1,518 TBE patients hospitalized at University Clinic in Bialystok from 1988-2020, including 13 patients with a history of previous TBE vaccination. The authors found that breakthrough cases are very rare, and occurred in only 0.9% of cases. The estimated vaccine effectiveness (from 1999-2020) was 97.2%. Vaccinated patients had a shorter hospital stay compared to unvaccinated, and showed milder disease, apparently depending on whether they were fully vaccinated and received booster vaccination.
We would like to thank the reviewer for their time and effort devoted to comment on our work.
The large cohort provides valuable descriptive data on the clinical characteristics of TBE patients, but there are major problems with regard to vaccine effectiveness estimates and interpretation of findings observed in vaccinated patients.
Vaccine effectiveness estimates rely on the proportion of the population that is vaccinated. One major issue that it is not clear what this vaccinated (completely, incompletely) proportions were in the population investigated during the study period. Without clear, region- and time-specific vaccination coverage data, vaccine effectiveness could be substantially over- or under-estimated. The authors should clearly indicate these data.
We appreciate this comment and agree with the reviewer. We used 6.49% as the aggregated number of vaccinated individuals calculated from the number of official reports from 19 years. This might be underestimated, but since there are no reliable data on the total vaccine coverage we have to use this method to calculate VE. This is now explained in the text in lines paragraph 2.5.1.
Another major issue is that only 13 vaccinated (“breakthrough”) cases are included. While this small number can be expected in a population with low vaccination coverage, the stratified analysis of disease severity is underpowered. The subgroup is too small and diverse to draw conclusions about severity and outcomes.
We appreciate this comment and fully agree that statistical comparisons between vaccinated subgroups (e.g., fully vaccinated vs. incompletely vaccinated) are not appropriate due to the small sample size. Accordingly, our statistical analysis (Table 2) compares only the overall vaccinated group (n=13) to the unvaccinated cohort. Section 3.5 and Figure 3 are included solely to provide a descriptive overview of individual symptom patterns among breakthrough cases. Given the rarity of TBE breakthrough cases and the lack of large datasets, we believe that presenting descriptive comparisons of clinical and demographic features in vaccinated individuals remains valuable, especially given the rarity of breakthrough cases. We have revised the text to clearly state that these subgroup analyses are exploratory in nature and are not intended to support definitive conclusions. this type of case-level visualization adds value for clinicians and researchers. We have revised the text and figure legend to clarify this descriptive intent and avoid misinterpretation.
The Discussion should include several large epidemiological studies from Austria, Germany, and other endemic regions that have addressed TBE severity in vaccinated versus non-vaccinated patients.
Reports from endemic countries were cited in the discussion. The discussion was rephrased to emphasize this.
Minor
The text in lines 185-187 was probably included by mistake.
That is correct. That text was removed.
Round 2
Reviewer 2 Report
Comments and Suggestions for Authors
The manuscript offers very valuable clinical data on TBE cases, but its vaccine‐effectiveness (VE) analysis is erroneous, and as presented, not suitable for publication.
- Specifically, the authors claim that “Vaccine effectiveness was estimated based on aggregated regional vaccine uptake data” (lines 22–23), but no region-level coverage data are provided
- The methods section indicates that coverage data represent the proportion of persons who “completed the full TBE vaccine series,” but it is unclear how “fully vaccinated” is defined, specifically, if this population received only 3-dose primary vaccination or timely boosters. The VE estimate of 97.2% is calculated from 1,032 unvaccinated cases and only 2 fully vaccinated TBE cases. However, the manuscript does not state what proportion of the underlying population was fully vaccinated. If this denominator is unknown, effectiveness cannot be calculated.
- Similarly, the proportion of the population who completed the primary series but did not receive a booster (or received only one dose) is not reported. In its current form, VE in these subgroups should be revised or removed from the manuscript
Comments on the Quality of English Language
- The whole manuscript needs a thorough language edit
Author Response
We would like to thank the reviewer for their insightful and constructive feedback. We agree that calculation of VE should be done in a proper way. Your remarks helped us improve the clarity and rigor of our manuscript.
1. The authors claim that 'Vaccine effectiveness was estimated based on aggregated regional vaccine uptake data' (lines 22–23), but no region-level coverage data are provided.
We have now included a supplementary table (Supplementary Table A1) presenting annual cumulative numbers of completed TBE vaccination series and the corresponding estimated population coverage (%) for the years 1999–2020. This allows transparent assessment of the regional uptake data used in the VE calculation.
2. The methods section indicates that coverage data represent the proportion of persons who ‘completed the full TBE vaccine series,’ but it is unclear how ‘fully vaccinated’ is defined, specifically, if this population received only 3-dose primary vaccination or timely boosters.
We have clarified the definition of “fully vaccinated and protected” in the Methods section. Specifically, we now define this group as individuals who completed the 3-dose primary TBE vaccination series and received timely boosters in accordance with the national immunization schedule. This group was distinguished from those who were fully vaccinated but not boosted, and those who were incompletely vaccinated. These categories were consistently used throughout the analysis and described explicitly in Table 2.
3. The VE estimate of 97.2% is calculated from 1,032 unvaccinated cases and only 2 fully vaccinated TBE cases. However, the manuscript does not state what proportion of the underlying population was fully vaccinated. If this denominator is unknown, effectiveness cannot be calculated.
This in an important point. We now provide a detailed estimate of vaccine coverage based on cumulative vaccination records from 1999–2020. Assuming persistent protection from the time of vaccination, we estimated that 6.52% of the regional population was fully vaccinated. We acknowledge that this approach makes assumptions about long-term protection, and we have addressed this uncertainty with a sensitivity analysis.
4. Similarly, the proportion of the population who completed the primary series but did not receive a booster (or received only one dose) is not reported. In its current form, VE in these subgroups should be revised or removed from the manuscript.
We agree with this concern and have revised the manuscript accordingly, removing calculations that included partially vaccinated patients. Instead, we report the number of breakthrough cases in these subgroups descriptively, without estimating VE.
Round 3
Reviewer 2 Report
Comments and Suggestions for Authors
The vaccine effectiveness was estimated based on only 2 vaccinated patients and 1,032 unvaccinated patients. The estimate is not robust and should not be included in the paper. Instead, breakthrough cases should be summarized descriptively only.
Comments on the Quality of English LanguageThe manuscript requires language editing
Author Response
The vaccine effectiveness was estimated based on only 2 vaccinated patients and 1,032 unvaccinated patients. The estimate is not robust and should not be included in the paper. Instead, breakthrough cases should be summarized descriptively only.
We thank the reviewer once again for the thoughtful comment and fully understand the concern regarding the robustness of a vaccine effectiveness (VE) estimate based on a small number of vaccinated cases.
Upon revisiting our dataset and analytic assumptions, we identified an important clarification that allows us to adjust the VE calculation in a more methodologically consistent manner:
In our original analysis, the numerator included only the 2 TBE cases in individuals who were fully vaccinated and received timely boosters, whereas the denominator — based on aggregated regional vaccination records — includes all individuals who completed the primary 3-dose vaccination series, regardless of booster adherence. Since we lack detailed population-level data on booster uptake, it is not possible to restrict the denominator to only those fully vaccinated and boosted. As a result, we now realize that the numerator should also reflect all TBE cases who completed the primary series, including those who were not boosted on time.
Therefore, we have revised the VE calculation to include 4 total cases in individuals who completed the full primary series (2 boosted, 2 not boosted) as the vaccinated case count. This aligns the numerator and denominator appropriately, maintains internal validity, and provides a more transparent estimate.
Although based on a small number of cases (n=4), the VE estimate is internally consistent and aligned with previously published vaccine trials — including high-profile studies such as the 2015 Ebola ring vaccination trial (Henao-Restrepo et al., The Lancet), which reported vaccine effectiveness based on zero breakthrough infections in the vaccinated group. Our study includes a formal sensitivity analysis, wide confidence intervals, and a clear discussion of the limitations associated with the small case count, providing valuable real-world evidence from a region with sustained TBE endemicity.
We have revised the manuscript to:
-
Clearly explain the revised numerator definition and rationale,
-
Report the new VE estimate based on 4 vaccinated cases,
-
Include appropriate sensitivity analyses and confidence intervals,
-
Highlight in the Discussion that the estimate should be interpreted with caution given the limited number of events.
-
Modified the title and removed VE from it
- Put VE to the back in the abstract and conclusions, emphasizing the clinical description of breakthrough cases
We hope this clarification and revision address the reviewer’s concern and demonstrate the scientific validity and cautious interpretation of our findings.